# Navigating the Multiverse of Antisense RNAs: The Transcription- and RNA-Dependent Dimension

**DOI:** 10.3390/ncrna8060074

**Published:** 2022-10-26

**Authors:** Giulia Pagani, Cecilia Pandini, Paolo Gandellini

**Affiliations:** Department of Biosciences, University of Milan, 20133 Milan, Italy

**Keywords:** antisense RNA (asRNA), long non-coding RNA (lncRNA), RNA-dependent mechanism, transcription-dependent mechanism, functional study

## Abstract

Evidence accumulated over the past decades shows that the number of identified antisense transcripts is continuously increasing, promoting them from transcriptional noise to real genes with specific functions. Indeed, recent studies have begun to unravel the complexity of the antisense RNA (asRNA) world, starting from the multidimensional mechanisms that they can exert in physiological and pathological conditions. In this review, we discuss the multiverse of the molecular functions of asRNAs, describing their action through transcription-dependent and RNA-dependent mechanisms. Then, we report the workflow and methodologies to study and functionally characterize single asRNA candidates.

## 1. Introduction

It has now become evident that a significative fraction of the transcriptome comprises RNAs that are transcribed in the antisense direction with respect to known genes. Despite being considered transcriptional noise during the past years, antisense transcription has now been recognized as one of the main mechanisms of gene expression regulation in all kingdoms of life [1]. This process produces numerous previously unknown antisense RNAs (asRNAs), also called natural antisense RNAs (NATs), belonging to the bigger family of the long non-coding RNA (lncRNA) and challenges our traditional definition of functional regions of the genome and genes. The produced transcripts are transcribed from the opposite strand to that of the sense transcript of either protein-coding or non-protein-coding genes. They are involved in almost all stages of gene expression regulation (from transcription to RNA degradation) and can function as a fast-evolving regulatory switch in creating transcriptional and cellular organismal complexity, either through the ncRNA sequence that is produced or the act of transcription itself [2,3,4].

Although antisense transcripts were discovered in bacteria more than 30 years ago [5], only the introduction of high-throughput in-depth transcriptome sequencing techniques, such as next generation sequencing, allowed us to understand that more than 30% of annotated transcripts in humans have antisense transcription [6]. This difficulty in annotating non-coding antisense transcripts is mostly due to their low abundance (they are, on average, 10-fold less expressed than sense transcripts) [6], thus making them challenging to study. Moreover, we still do not know the exact extent of antisense transcription in the human genome. As a matter of fact, the Telomere-to-Telomere Consortium recently presented a complete 3055 billion-base pair sequence of the human genome, T2T-CHM13, addressing the remaining 8% of the genome [7]. Nearly 200 million base pairs of novel sequences were identified, accounting for 1956 gene predictions. However, only 99 of them are predicted to be protein-coding, indicating that we are just at the beginning of understanding the non-coding part of the genome.

Functional studies showed that antisense transcripts are not a uniform category of regulatory RNAs but comprise different groups of RNAs sharing some common features. A possible way to categorize them is by their position and relationship with their sense gene (Figure 1). First of all, antisense transcripts can overlap or not with their sense gene. Overlapping asRNAs are convergently transcribed with respect to the sense gene and can be classified into (a) head-to-head transcripts, when the antisense and the sense RNA share complementarity at the 5′ end; (b) tail-to-tail transcripts that share the 3′ end; (c) intronic transcripts, when they are transcribed from the intron of the sense gene; and finally (d) antisense transcripts can be completely embedded in the protein-coding transcript. Non-overlapping antisense transcripts are divergently transcribed with respect to their sense gene and localize in close proximity to it; (e) they can share the same promoter as the sense gene (bidirectional promoter) or (f) have an independent promoter. The position on the genome and the extent of complementarity with the sense gene can suggest the possible mechanism of action: in cis, if the asRNA acts on its sense gene at its genomic locus, or in trans, if the asRNA acts far from its locus. Moreover, asRNAs can sustain both gene inhibition and gene activation in several different ways, also depending on their nuclear or cytoplasmic localization.

In metazoan cells, asRNAs (and more in general, lncRNAs) exert multiple physiological roles (i.e., from cell cycle regulation [8] to circadian clock oscillatory mechanisms [9]) but they are emerging also as key players in different human diseases [10,11,12]. Most of the examples come from the cancer field, where the list of involved asRNAs is constantly growing. Notably, these lncRNAs have pivotal roles in gene regulation and they have been implicated in the acquisition of almost every hallmark of cancer, from the intrinsic capacity of proliferation and survival, through the increased metabolism, to the relationship with the tumor microenvironment, by acting either as oncogenes or tumor suppressors [13,14]. Moreover, asRNAs are involved in almost all other types of pathologies, from the most widespread neurodegenerative diseases (Amyotrophic Lateral Sclerosis [15], Alzheimer’s disease (AD) [16] and Parkinson’s disease [17]), to metabolic [18] and cardiovascular diseases [19]. Despite the molecular function of some asRNAs in diseases being quite well established, the mechanisms of action of most asRNAs are far from being fully understood. This is primarily linked to the complexity of the underlying phenomena, especially now that it is known that asRNAs can work in more than one mode of action.

In this review, we discuss the molecular dimensions of asRNAs dividing their mode of action into transcription-dependent mechanisms, where the antisense transcription itself is a regulatory process without the intervention of sequence-driven effects, and into RNA-dependent mechanisms, where asRNAs act due to their specific RNA sequence or structure. It is noteworthy that such modes of action are not mutually exclusive, as there are examples of asRNAs exploiting both modalities. We will then discuss the methodologies to discriminate these types of mechanisms of action and to study asRNAs at the molecular level.

## 2. Transcription-Dependent Mechanisms

It is now established that eukaryotes express many more RNA molecules with a biological function than what was expected according to the classical transcriptome view [20,21]. As described in Struhl et al. [22], only 10% of the elongating RNA polymerase II molecules in yeast are involved in transcription initiating at conventional promoters. Regarding the remaining 90%, the relative proportion of biologically relevant transcription and transcriptional noise is unknown. Indeed, transcription can also start at cryptic promoters located in many coding genes or within intergenic regions [22]. Such transcription events may occur on both strands, consequently originating the antisense and other non-coding RNAs present in eukaryotes [21]. However, a critical step is distinguishing functional transcription from transcriptional noise as both types of transcripts can be produced by RNA polymerase II machinery. In this regard, although evolutionary conservation can help identify some biologically relevant asRNAs, especially when their function directly relies on the sequence, it might be misleading when the biological function is related to transcription rather than the RNA product [22].

In this section we provide general concepts related to antisense transcription and describe antisense RNAs with a transcription-dependent mechanism, which are part of the bigger group of nuclear in cis-acting asRNAs [23], distinguishing them as non-overlapping and overlapping antisense RNAs.

### 2.1. Antisense Transcription

It is known that eukaryotic genes pervasively undergo non-coding transcription of their antisense strand [24]. In addition, a class of divergently transcribed gene pairs (sense and antisense), whose transcription start sites (TSSs) are less than 1000 bp distant, has been identified. Such gene pairs accounts for more than 10% of the genes in the genome and can be co-expressed, but also anti-regulated [25]. There is also evidence that the transcription of most asRNAs originates from bidirectional promoters of active protein-coding genes and may provide a new mechanism of regulation to mediate the response of the sense promoter to environmental changes. For example, many divergently transcribed antisense/sense gene pairs are coordinately regulated during mouse embryonic stem cell differentiation [26]. Moreover, antisense transcripts may be involved in pluripotency maintenance in mouse embryonic stem cells [27]. Accordingly, genes with indication of antisense transcription in yeast showed bigger variation in sense transcription, meaning that they are more transcriptionally plastic. In addition, genes that must respond in a switch-like manner, such as stress-response and environment-specific genes, are enriched for antisense transcription. Thus, antisense expression initiated from bidirectional promoters may allow the propagation of regulatory signals from one locus to nearby genes [28].

Moreover, antisense transcription is associated with distinct chromatin features, leading to a difference in sense transcription between genes with high or low antisense transcription levels [2,29,30]. In this regard, using a genome-wide approach, Murray et al. [29] found antisense transcription to be associated with a dynamic chromatin structure, and increased nucleosome occupancy, level of acetylation, and histone turnover across the promoter and body of genes, without necessarily modulating the sense transcription level. As a confirmation, using yeast mutants in which antisense transcription across the *GAL1* gene is reduced, the loss of antisense transcription causes expected changes to the chromatin [29]. According to this, antisense transcription influences sense transcription dynamics in a chromatin-dependent manner. Brown et al. [30] developed a stochastic model to describe transcription from initiation to transcript degradation and found that increased levels of antisense transcription alter sense transcription dynamics, reducing transcript production and processing rates, while increasing transcript stability. Importantly, these changes in transcription dynamics are directly influenced by antisense transcription in a chromatin-dependent manner. Indeed, the disruption of the Set3 histone deacetylase activity establishes antisense transcription-associated chromatin signature and recapitulates the effect of antisense transcription on sense transcription dynamics [30].

Therefore, by influencing the chromatin architecture, antisense transcription should be considered an important regulatory mechanism of eukaryotic genes, which may make sense genes more sensitive to signals and prime them for responses to developmental issues or stressful environments [2].

### 2.2. Non-Overlapping asRNAs

In the case of non-overlapping asRNAs, sense and antisense genes are divergently transcribed and can act at the transcription initiation level by altering RNA polymerase II occupancy on neighboring promoters (Figure 2a). An example is *linc1319* or *Blustr* (bivalent locus (*Sfmbt2*) is upregulated by splicing and transcribing an RNA). Both promoter deletion and insertion of polyadenylation signals (PASs), which induces premature transcription termination at different lengths, were shown to result in a decrease of *Sfmbt2* expression, a nearby gene located 5 kb upstream. Such a cis-activating effect is independent of *Blustr* sequence, as sequential deletions of downstream exons and introns do not impair *Sfmbt2* expression; instead, it is correlated with transcription length. Additionally, impaired *Blustr* transcription affects the chromatin state of the *Sfmbt2* promoter and decreases RNA polymerase occupancy at the transcription start site and within the gene body of *Sfmbt2* [31]. *Upperhand* (*Uph*) is an asRNA divergently transcribed from a bidirectional promoter that also produces *Hand2*, a heart development regulator. Although the *Uph* sequence is not much conserved in different species, production of a transcript from this locus spanning multiple well-conserved *Hand2* enhancers is common across mammals. Insertion of PASs leading to premature transcription termination of *Uph* impairs *Hand2* expression in cis, resulting in embryonic lethality in mouse. Instead, the insertion of a heterologous DNA sequence at the same site does not disrupt *Hand2* expression. Accordingly, knockdown of the *Uph* mature transcript through Gapmer antisense oligonucleotides (ASOs) does not affect *Hand2* expression. Specifically in this example, premature transcription termination reduces the enhancer chromatin signature at the *Uph* gene body and prevents binding of the transcription factor GATA4 to a *Hand2* enhancer in this region, thus resulting in decreased *Hand2* expression [32].

### 2.3. Overlapping asRNAs

On the other hand, in cases of overlapping asRNAs, sense and antisense genes are convergently transcribed. Also in this case, they can regulate transcription initiation by transcriptional interference (Figure 2b). Indeed, the assembly of the transcriptional machinery at one promoter can physically prevent the assembly at the other promoter (promoter competition) [33]. The passage of the RNA polymerase can occlude the binding sites for the other RNA polymerase (promoter occlusion) [34]. An example is how the paternally expressed *Airn* (antisense to *Igf2r* (insulin-like growth factor type 2 receptor) RNA) silences in cis the paternal alleles of *Igf2r* in the *Igf2r* imprinted cluster. Different shortened *Airn* transcripts created by insertion of PASs do not affect *Igf2r* silencing in mice. Instead, the transcriptional overlap with the *Igf2r* promoter interferes with the recruitment of RNA polymerase II, leading to *Igf2r* silencing [35]. Transcriptional interference can be particularly important in cases of sense–antisense pairs that have a role as core clock genes, allowing their rhythmic and antiphasic expression pattern. Recently, Mosig et al. [9] demonstrated that *Per2AS,* the asRNA of *Period2* gene, regulates its sense gene exclusively through transcriptional interference, resulting in a negative feedback loop.

Antisense expression can also regulate gene expression after transcription initiation by transcriptional interference that occurs co-transcriptionally (Figure 2c). This can be mediated by transcriptional interference due to direct collision caused by the concomitant and convergent progression of bulky transcription complexes on opposite strands [36]. In this regard, transcriptional interference by collision occurs most likely in cases of two strong convergent transcriptional complexes; instead, it is less likely for weak transcriptional complexes to be simultaneously transcribed. However, polymerase pausing can extend the polymerase occupancy time so that it may increase transcriptional interference, also for weak promoters [37]. Other mechanisms include RNA polymerase acting as an obstacle or roadblock for other incoming elongating polymerases, or the sitting-duck interference, when an elongating polymerase removes another one already bound to its promoter [38,39]. An example of transcriptional interference acting co-transcriptionally is the antisense-mediated repression of *IME4*, encoding a meiosis key regulator, in budding yeast [40,41]. In diploid cells, the α1-α2 complex binds downstream of *IME4* and represses the transcription of the overlapping antisense *RME2* (regulator of meiosis 2), allowing *IME4* to be induced during meiosis. Instead, in haploid cells the expression of *RME2* represses *IME4*. Accordingly, mutation in the α1-α2 binding site allows expression of *RME2* and prevents *IME4* expression in diploid cells. Moreover, promoter deletion of *RME2* in haploid cells leads to an absence of *RME2* expression and *IME4* derepression. In addition, premature termination of *RME2* transcription obtained by insertion of PASs alters its ability to repress *IME4* in haploid cells. Concordant with this, deletion analyses reveal that a precisely oriented sequence within the *IME4* ORF is necessary for *RME2*-mediated repression. Indeed, when such a region is deleted, *IME4* transcription is derepressed in haploid cells. This suggests that, through transcriptional interference, *RME2* transcription blocks polymerase elongation, but not initiation, at the *IME4* locus [41]. *nAS25* is the asRNA encoded in exon 25 of the *notch-1* locus [8]. When it is expressed by a promoter bound by the E2F1 transcription factor in the G1 phase, it causes the interruption of full-length *notch-1* transcription, defining a slow harmonic oscillation of *notch-1* and *nAS25* and coordinating cell-cycle dynamics. This event occurs probably due to the convergence of polymerization by RNA polymerase from template and non-template DNA strands. Indeed, in the absence of antisense transcription, processivity of RNA polymerase in the sense direction increases as there is a reduction of positive supercoiling of DNA.

In particular, the two examples of *Per2AS* and nAS25 are emblematic of the complexity and multilayered activities of asRNAs. Indeed, in addition to a transcription-level impact, they also show a transcript-level function by interacting with other targets (such as *Bmal1* for *Per2AS*) [9] or even with the same sense gene (*notch-1* for *nAS25*) [8] through RNA-dependent mechanisms (described in the next section). The combination of these two aspects is common to many other asRNAs, resulting in a fine-tuned regulatory process of cell life.

## 3. RNA-Dependent Mechanisms

The most convincing demonstration that asRNAs are not only transcriptional noise, but real genes with specific functions, is derived from the discoveries of mechanisms of action based on their specific RNA sequences and/or structures (Figure 3). Indeed, asRNAs have the peculiar feature that they bind to their targets due to sequence complementarity. In most of the cases, the complementarity is with the sense gene; thus, the asRNA acts in cis at its genomic locus, but there are also some examples of asRNAs that work in trans pairing with mRNAs from other genes in the nucleus or in the cytoplasm, for example, by sponging microRNAs (miRNAs). In cis and in trans modalities can also rely on other types of interactions, such as that with the DNA or proteins (e.g., splicing and epigenetic factors). This is made possible by the formation of complex secondary and tertiary RNA structures, including hairpins, stem-loops, bulges and duplex-, triplex- and quadruplex-motifs [42,43]. Since predicting the RNA structure from the asRNA sequence is challenging, understanding how asRNAs interact with DNA and proteins is not trivial [44]. However, multiple alignment analysis of lncRNAs has been performed and found short patches of conserved bases with a certain degree of conservation of secondary and higher-order structures [45,46,47]. This evidence paved the way for the hypothesis that lncRNAs, and thus also asRNAs, may work through discrete functional RNA modules or domains [44].

### 3.1. In Cis asRNAs

In cis asRNAs can act within their genomic locus with multiple mechanisms directly binding to DNA, RNA or proteins.

First of all, it is straightforward that asRNAs may use the classical Watson–Crick base pairing to bind to complementary RNA/DNA sequences. Formation of the RNA–RNA duplex can stabilize the target mRNA by preventing RNase-mediated degradation or by influencing the binding of RNA binding proteins (RBPs) to mRNA, as was proposed for *PDCD4-AS1* and *PDCD4* in breast cancer [48] and for *FOXC2-AS1* and *FOXC2* in colorectal cancer progression [49], respectively (Figure 3a). Another example is *nAS25* asRNA which provides partial protection from RNA editing for *notch-1* transcript by hybridizing to it and acting as a decoy target for the C➔U editing enzyme APOBEC-1 (Figure 3a). The peculiarity of this case is that this in cis base pairing mechanism is strictly temporally controlled and happens in daughter cells that have inherited *nAS25* from mother cells, thus ultimately establishing an oscillatory system that coordinates cell cycle duration [8].

Antisense expression can also regulate the production of alternative transcript isoforms of the sense gene (Figure 3b). In this regard, the antisense transcript can mask specific splice sites present in sense mRNA and thus influence its alternative splicing or the consequent protein translation. One example of alternative isoform production has been found in AD where the asRNA *51A* maps to the first intron of Sortilin-related receptor 1 (*SORL1*) and, by pairing with its pre-mRNA, drives a splicing shift of *SORL1* from the canonical full-length protein form to an alternatively spliced shorter protein variant that is associated with an impaired processing of the amyloid-precursor protein (APP) [50]. asRNA–mRNA interaction can also mask splicing sites, preventing their processing, thus resulting in remodulation of protein translation. In humans, the zinc-finger E-box-binding homeobox 2 gene (*ZEB2*) encodes for a transcriptional repressor of E-cadherin. Its antisense transcript *ZEB2-AS1* hinders the processing of an intron in the 5′UTR, which harbors an internal ribosome entry site (IRES) on the *ZEB2* mRNA, resulting in increased protein production [51]. Moreover, the asRNA–mRNA duplex can undergo deamination of adenosine to inosines (Figure 3a), which can result in nuclear retention or cytoplasmic degradation [52].

Different asRNAs have been reported to interact with DNA by the establishment of R-loops which form when a G-rich RNA transcript invades the DNA duplex, annealing to the complementary strand and displacing the G-rich DNA strand to generate an RNA–DNA hybrid structure (Figure 3d). This mechanism was described for *VIM-AS1*, which regulates its sense *VIM* gene through the formation of an R-loop that is crucial to creating the optimal chromatin environment to favor the binding of transcription factors [53].

Another way by which asRNAs can interact with other molecules is through their spatial conformation. Together with the DNA, asRNA can originate a triplex, in which a double-stranded DNA forms a triple helical structure by accommodating a single-stranded RNA in its major groove through Hoogsteen or reverse Hoogsteen base pairing [4] (Figure 3f). Upon triplex formation, asRNAs may recruit protein complexes at specific genomic sites to regulate gene expression [54]. In the mouse, a 20 nt-long segment of *Fendrr* asRNA (also known as *FOXF1-AS1*) is predicted to form a triple-helix with a complementary sequence within the *FOXF1* promoter, thus attenuating *FOXF1* expression [55]. At the epigenetic level, in cis asRNAs can indeed repress transcription by causing to interact and by recruiting histone and chromatin remodeling proteins to their genomic locus (Figure 3e). Some of the first examples of this type of mechanism were observed with asRNAs of tumor suppressor genes [56,57]. These asRNAs were observed to epigenetically alter the chromatin to a more compact state, thus modulating the activity of the promoter of the targeted protein-coding gene. It is worth noting that some level of asRNA expression has been found to be associated with most of the tumor suppressor genes reported to date. These tumor suppressor genes are also often the ones found to be epigenetically silenced in human cancers [56]. For instance, the antisense *AS1DHRS4* was shown to negatively regulate the expression of its sense *DHRS4* in cis exerting more than one of the mechanisms listed above. *AS1DHRS4* forms an RNA duplex with its nascent sense transcript to mediate repressive histone modifications; moreover, *AS1DHRS4* was also shown to interact with two histone methyltransferases to keep epigenetic silencing of *DHRS4* locus and to recruit DNA methyltransferase to induce DNA methylation in the promoter region [52].

### 3.2. In Trans asRNAs

In trans-acting asRNAs are an expanding class of lncRNAs that leave their site of transcription to act both in the nuclear and in the cytoplasmic compartment. In trans effects can be mediated by antisense transcripts that interact with distant loci through the three-dimensional organization of chromatin and by the mechanisms of interaction listed above for in cis asRNAs (but in this case the target locus is not the asRNA originating locus). The existence of in trans functions somehow raises a nomenclature issue for asRNAs, which will be examined in the Conclusions section. Indeed, some of the so-called asRNAs are only casually transcribed in the opposite direction with respect to a given sense gene, without regulating it in cis or being the result of the same act of transcription [58]. In contrast, some lncRNAs not classified as asRNAs, act on other RNAs by sequence complementarity, thus exerting an antisense function (an example can be found in [59]).

Base pairing mechanisms are used also by in trans asRNAs to regulate the abundance or activity of other RNAs (usually miRNAs, but also mRNAs) far from their genomic locus, usually by sequestration or stabilization [23]. The asRNA Ubiquitin C-Terminal Hydrolase L1 *(Uchl1)-AS1* is an example of asRNA–mRNA interaction. It is predominantly nuclear, but it can shuttle to the cytoplasm under stress conditions to bind to the 5′UTR of *Uchl1* mRNA, thus triggering increased translation of the UCHL1 protein [60] (Figure 3a). β-site amyloid precursor protein (APP)-cleaving enzyme 1 antisense (*BACE1-AS*) is the antisense of BACE1, a crucial enzyme in the pathophysiology of AD which can lead to the formation of the APP plaques. It enhances the stability of the *BACE1* mRNA by binding to a region of partial complementarity, masking a site for *miR-485-5p*, which would otherwise suppress BACE1 biosynthesis [61] (Figure 3a). Moreover, the RNA duplex can also be a substrate of the Dicer protein in the cytoplasmic RNA interference (RNAi) pathway (Figure 3a), eventually leading to degradation or translation repression of the sense mRNA [62]. Curiously, in the cited examples, asRNAs ultimately regulate their sense gene, but through cytoplasmic in trans mechanisms based on sequence complementarity rather than locally in cis at the transcriptional/epigenetic level.

Finally, in trans-acting asRNAs can act as competing endogenous RNAs (ceRNA) sponging miRNAs (Figure 3c). There are a growing number of examples of this mechanism [63,64,65,66], which is based on a competitive asRNA-miRNA binding that ultimately sequesters miRNAs from their original target transcripts, thus avoiding the degradation or translation inhibition of the target.

On the other hand, in trans antisense transcripts can physically interact with DNA and proteins in a sequence-independent manner. Indeed, the mouse asRNA *Fendrr*, which, as stated above, acts in cis forming a RNA–DNA triplex with its sense gene (Figure 3f), can exert exactly the same mechanism to regulate in trans at least another gene, *PITX2* [55]. One of the most known examples is HOX transcript asRNA (*HOTAIR*) in mammals, which is the antisense transcript to the homeobox C (*HOXC*) locus. Through a 5′ domain, *HOTAIR* binds to the Polycomb Repressive Complex 2 (PRC2), which induces histone H3 lysine 27 trimethylation, thus silencing *HOXD* locus in trans [67] (Figure 3e). *HOTAIR* is not the only asRNA that can exert its function in trans by interacting with PRC2; in fact, guidance of PRC2 by antisense transcripts is likely to be common, as the complex was shown to directly interact with more than 3000 asRNAs [68]. *HOTAIR* also functions as a molecular scaffold for LSD1, which is involved in demethylation of histone H3 at lysine 4. *HOTAIR* binds to LSD1 through a 3′ domain coordinating its function in chromatin modification [69]. *BACE1-AS* is also able to bind to the cytoplasmic RBP HuD (ELAVL4) (Figure 3a), which increases the stability of *BACE1-AS* itself as well as that of *BACE1* and *APP* mRNAs, thereby coordinating several steps that converge on the generation of amyloidogenic plaques in AD brain [70].

## 4. Methodologies to Discriminate and Study asRNA Mechanisms of Action

### 4.1. Identification of asRNAs

Thanks to recent advances in high-throughput and in-depth transcriptome sequencing technologies, RNA-seq is widely used for the identification of novel asRNAs [71,72]. As asRNAs can be both polyadenylated and non-polyadenylated RNAs, total RNA-seq should be the best choice to obtain informative results [73]. Moreover, with the increasing use of single-cell sequencing, RNA-seq becomes one of the most powerful approaches to identify asRNAs at a single cell level [73,74]. RNA immunoprecipitation is the other most widely used method to isolate asRNAs interacting with a specific protein of interest [73].

Given that the antisense and sense genes can overlap to different extents, a precise and correct annotation of both genes is a key step in the study of the asRNA. By exploiting different high-throughput sequencing data such as cap analysis of gene expression (CAGE-seq) [75,76], 3′ end-sequencing (3′ end-seq) [77,78], and by experimentally performing 3′ and 5′ rapid amplification of cDNA ends (RACE) experiments [79,80], it is possible to define the correct annotation of the sense and antisense genes in terms of TSS and transcription termination site (TTS). Long read sequencing data may then be useful to further define exon composition and alternative splicing patterns. Moreover, it is important to investigate the regulatory regions of both genes, such as promoters and enhancers, by taking advantage of publicly available data of histone modifications [81], DNAse sensitivity signal [82] and ChIP-seq peaks [83], to start to get hints about possible co-regulation.

### 4.2. Functional Study of asRNAs

Once a putative asRNA of interest is identified, the dissection of its possible mode of action is not trivial. It is important to discriminate whether the asRNA acts locally in cis, regulating the expression of the sense gene or nearby genes, or acts in trans, regulating the expression of distant genes. To distinguish all the possible modes of action, different loss- and gain-of-function strategies should be employed, as described in detail by Kopp et al. [84]. Here, we briefly describe and highlight the main aspects (Figure 4).

A primary characterization is the study of the subcellular localization of the asRNA, which can be assessed by cell fractionation or immunofluorescence. This can be helpful to get hints concerning its nuclear function in cis or on its nuclear or cytoplasmic function in trans.

To discriminate an in cis- from an in trans-acting function, it is possible to employ CRISPR/Cas9-based strategies to perform asRNA gain- and loss-of-function. For example, the deletion of the promoter or the entire locus can be exploited to abrogate the endogenous antisense transcription. The insertion of a PAS near the 5′ end of the asRNA locus can be used to induce premature transcription termination. CRISPR activation (CRISPRa) or inhibition (CRISPRi) can be employed to either activate or repress the endogenous transcription of the asRNA. In this regard, knowing the exact structure of the sense and antisense gene is crucial to design and adapt the strategy. Indeed, when the endogenous transcription of an antisense gene is manipulated to verify how this impacts on the expression of the sense gene, it is essential to exclude the fact that the used strategy may itself directly modulate the expression of the latter, for example, when relevant promoter/enhancer regions are edited. Afterward, expression/transcription of nearby and distant genes should be studied, as can be measured through quantitative PCR or whole transcriptomic analysis. In addition, as these strategies affect the endogenous asRNA transcription, nascent transcript analysis and ChIP to monitor polymerase occupancy can be performed to better evaluate the effect of these strategies on antisense transcription and consequently on sense and nearby gene transcription. In cases of an in cis-acting asRNA, an altered expression/transcription of such genes is expected. Moreover, the ectopic overexpression of the asRNA in knockout cells will not rescue the phenotype observed after antisense loss-of-function, as the asRNA will not be transcribed at its locus, thus further confirming its action in cis. In this case, the act of transcription of the asRNA is necessary for its activity or the RNA molecule may perform a regulatory function in a sequence/structure-specific manner. To discriminate transcription- and RNA-dependent mechanisms of action of in cis-acting asRNAs, it is possible to systematically introduce targeted mutations that disrupt splice sites and sequentially delete or replace sequences of the asRNA (such as exons). For instance, an asRNA that functions in a sequence-specific manner will be sensitive to removal of exons, while loci in which transcription is the key event will often tolerate sequence substitutions but will be inactivated by PAS insertion.

On the other hand, in case of an in trans-acting asRNAs, an altered expression of distant genes is expected. In this case, the function in trans should be rescued by the ectopic overexpression of the asRNA in knockout cells. Indeed, the phenotype resulting solely from the ectopic overexpression of the asRNA is exclusively due to its function in trans. Inducible systems can be used to control expression/editing of asRNAs temporally and spatially. Vectors with Tetracyclin-/Cumate-/Tamoxifen-controlled system are some of the classical and most used tools [85]. More recently, other types of inducible system were developed as light-sensitive liposome delivery [86] and conditional Cas9 destabilization trough the fusion of FKBP12-derived destabilizing domain to Cas9 [87].

In parallel, to study whether an asRNA locus produces a functional RNA molecule, ASOs or short interfering RNAs (siRNAs) can be used to knockdown the expression of the mature antisense transcript, as both siRNAs and ASOs target complementary RNA molecules. siRNAs can be employed to study RNA-dependent mechanisms in the cytoplasm as they exploit the RNAi pathway, which results in the cleavage of complementary target RNA in the cytoplasm [88]. In contrast, ASOs are usually employed to study nuclear RNA-dependent mechanisms, as they can also function in the nucleus by slicing or physically blocking the complementary target RNA [89]. In cases where ASO-mediated knockdown of a given asRNA alters the expression of neighboring genes, it is likely to work in cis through RNA-dependent mechanisms, though additional experiments should be performed to assess eventual transcription-dependent functions (discussed below). Otherwise, if the expression of distant genes is altered, a function in trans becomes more likely.

Regarding ASOs, it can be useful to consider the use of Gapmers, a chimeric class of ASOs. The central part of the Gapmer is a DNA antisense oligonucleotide able to bind to the complementary target RNA, thus forming an RNA–DNA-like duplex that induces the RNase-H-mediated cleavage of the target RNA. Instead, the two flanking regions of the Gapmer are locked nucleic acids (LNAs). The LNAs consist of special nucleotides in which the ribose moiety is modified with an extra bridge connecting the 2′ oxygen and the 4′ carbon forming a locked structure. This structure confers increased stability against enzymatic degradation and increased specificity and affinity for the complementary target RNA [89].

It has been recently demonstrated that Gapmers can also target nascent transcripts and, when directed against regions in the proximity of the TSS of the target RNA, can trigger premature transcription inhibition [90]. Indeed, the endonucleolytic cleavage of nascent transcripts either by the polyadenylation machinery or through other mechanisms is a natural trigger for transcription termination. After the cleavage, the resulting uncapped 5′ end becomes a substrate for the 5′-to-3′ exoribonuclease XRN2, which degrades the nascent transcript and eventually reaches the elongating polymerase II complex, inducing the displacement of such complex [91]. Therefore, Gapmers proximal to the TSS of an asRNA may also affect its transcription, while Gapmers distal from its TSS are expected to cleave the RNA molecule without affecting transcription. Thus, it is possible to use both proximal and distal Gapmers to explore RNA and transcription-dependent mechanisms. Notably, light-sensitive liposome delivery [86] may be used also for the temporally controlled release of synthetic ASOs or siRNAs.

Finally, the identification and study of proteins or nucleic acids (DNA and RNA) that interact with the asRNA is the next step in the functional study of RNA-dependent modes of action. Concerning this, different pulldown strategies can be exploited. Chromatin RNA immunoprecipitation (ChIRP) is used to study the RNA–DNA interactions [92], RNA antisense purification (RAP)–RNA for the RNA–RNA interactions [93] and RNA pulldown for RNA–protein interactions [94]. In addition, different derivations of RNA immunoprecipitation can be very useful to investigate whether the asRNA under study interacts with a specific protein. For example, native RNA immunoprecipitation (RIP) is a powerful technique used to detect interactions between a specific protein and RNA molecules in vivo [95]. Cross-linking and immunoprecipitation (CLIP) combines immunoprecipitation and UV cross-linking to identify the RNA sites bound by a specific RNA-binding protein [96].

## 5. Conclusions

The number of identified antisense transcripts is continuously increasing and the role of antisense transcription as an intrinsic mechanism for regulation of transcription itself is starting to be acknowledged and understood. However, the roles of antisense transcripts go far beyond the in cis regulation of sense genes through their act of transcription. Several examples of asRNAs that do not regulate their sense gene in cis, as well as asRNAs that are the result of independent transcriptional events (i.e., transcribed from independent promoters), have indeed been reported in the literature. In addition, there is plenty of evidence of asRNAs that regulate their genes in cis through RNA-dependent mechanisms that rely on their sequence or structure, rather than through transcription-mediated events. Even more strikingly, the number of transcripts that, though being annotated as asRNAs, do not regulate their sense genes but control other genes in trans in the nucleus or in the cytoplasm is continuously growing. Among these, only a fraction of asRNAs (as for positional definition) truly act as antisense of, i.e., pair to, other RNAs. In many cases, gene regulation occurs via the formation of complexes with the DNA and/or proteins.

In view of this evidence, the definition of asRNA should be at least debated in the scientific community to distinguish RNAs that are the result of antisense transcription from a bidirectional promoter (transcription-based definition) and RNAs that work as antisense to other RNAs (mechanism-based definition). Though the two aspects can come together for asRNAs that are transcribed and overlap with their sense gene (e.g., *FOXC2-AS1* and *FOXC2* [49]), most RNAs of the first class regulate their genes in cis by transcription-dependent mechanisms that do not rely on an antisense pairing, whereas most of the latter use other RNA-dependent mechanisms to regulate genes that can be very distant from their genomic location or even mRNAs and miRNAs in the cytoplasm. As mentioned, a number of genes annotated as antisense for proximity with other genes do not fall in either of the two classes, as they may not be transcribed together nor regulate their gene in cis. Our view is that such RNAs should be reannotated as independent lncRNAs with their own unique name and regulatory function. In the absence of experimental data proving an action in cis, at least a certain window of proximity should be defined to annotate a novel RNA that is transcribed in the antisense direction to a known sense gene (without overlap) as antisense to it or not.

Aside from this lack of uniform classification and annotation, it is clear that asRNAs represent a multiverse of regulatory RNAs, the functions and mechanisms of action of which are very far from being completely understood. In this review, we reported the approaches that can be used to identify and functionally characterize novel asRNAs (Figure 4). Specifically, we described the potential of different manipulation approaches, mainly CRISPR/Cas9- and ASO-based methods, to assess transcription- vs. RNA-dependent mechanisms of action, as well as in cis vs. in trans function of asRNAs.

In addition to it being interesting to understand more of the complexity of gene regulation, asRNAs are a rich environment for the discovery of novel biomarkers and therapeutic targets, in light of their deregulation in different human pathologies. For example, *HOTAIR* has been shown to be a factor of both metastasis and survival in primary breast tumors [97]. *BACE1-AS*, which exerts a fundamental role in AD pathogenesis, can be detected in the plasma of AD patients and thus serves as a potential disease biomarker [16,98]. Moreover, in vitro manipulation of disease-relevant asRNAs has been proven to rescue pathological phenotypes, which would anticipate a future therapeutic applicability. For instance, the knockdown of asRNA overexpressed in tumors can impact on different hallmarks of cancer (i.e., proliferation, metastasis, epithelial-mesenchymal transition), as described for *ZEB2-AS1* [51] and *FOXC2-AS1* [49]. In this regard, several intrinsic characteristics of asRNAs, such as lack of translation, low expression, rapid turnover and high tissue-specificity, represent theoretical clinical advantages for their use as therapeutic targets. In addition, the same approaches used for their functional dissection in vitro may be adapted for use in in vivo models, up to a possible clinical application in humans, provided that an accurate refinement of these methods is conducted in terms of pharmacological and safety profiles. In this regard, there are different examples of ASOs used in clinical trials, as nicely reviewed in Roberts et al. [99].

Overall, it is now evident that asRNAs are more than just the result of an act of non-coding transcription of the antisense strand that impacts locally on the expression of genes in cis. They are indeed a heterogeneous class of regulatory RNAs that exert specific functions in a variety of biological processes, through mechanisms that can also be dependent on their sequence/structure as well as alter the expression of distant genes in trans.

## Figures and Tables

**Figure 1 ncrna-08-00074-f001:**
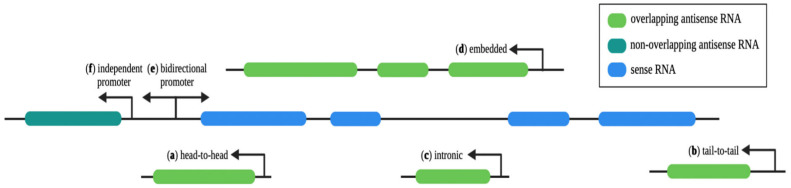
asRNA classification based on their genomic position and direction of transcription with respect to the sense genes. asRNAs are divided into two major categories: convergent overlapping asRNAs, which include (**a**) head-to-head, (**b**) tail-to-tail, (**c**) intronic and (**d**) embedded transcripts, and divergent non-overlapping asRNAs, which can (**e**) share a bidirectional promoter with their sense gene or (**f**) have an independent one. Created with BioRender.com accessed on 25 October 2022.

**Figure 2 ncrna-08-00074-f002:**
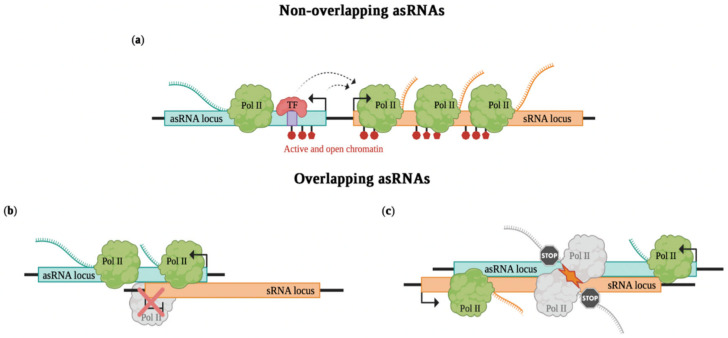
Transcription-dependent mechanisms of asRNAs. In the figure, the transcription-dependent modes of action of asRNAs are reported. In cases of non-overlapping asRNAs (**a**), the transcription or the binding of a transcription factor (TF) at the asRNA locus promote the transcription at the sense RNA (sRNA) locus. In cases of overlapping asRNAs (**b**,**c**), the transcriptional overlap (**b**) interferes with the recruitment of the RNA polymerase II (Pol II), causing sRNA locus silencing, or (**c**) leads to collision of the concomitant and convergent progression of bulky Pol II on opposite strands. Created with BioRender.com accessed on 25 October 2022.

**Figure 3 ncrna-08-00074-f003:**
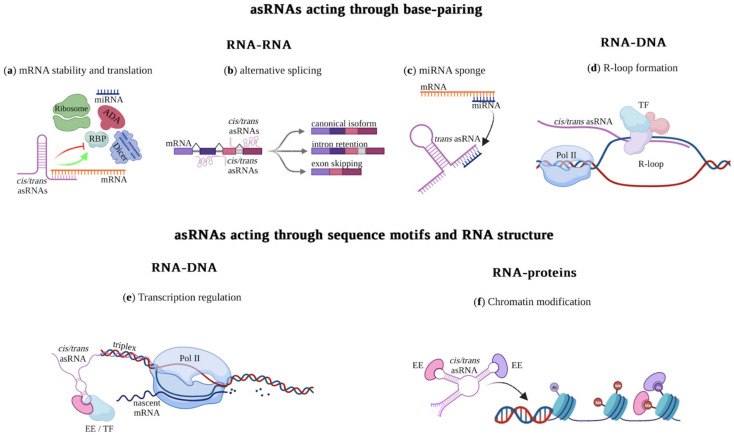
RNA-dependent mechanisms of asRNAs. In the figure, the different RNA-dependent modalities of asRNAs are depicted. asRNAs can act through base pairing with other RNAs by affecting the (**a**) mRNA stability and translation or (**b**) the alternative splicing of target genes; (**c**) they can also act as sponges to sequester miRNAs or (**d**) form R-loop with the DNA helix. On the other hand, asRNAs can act through sequence motifs and RNA structure, allowing them to bind to (**e**) DNA and (**f**) proteins, respectively. ADA: adenosine deaminase; RBP: RNA binding protein; miRNA: microRNA; Pol II: RNA Polymerase II; TF: transcription factor; EE: epigenetic enzyme. Created with BioRender.com accessed on 25 October 2022.

**Figure 4 ncrna-08-00074-f004:**
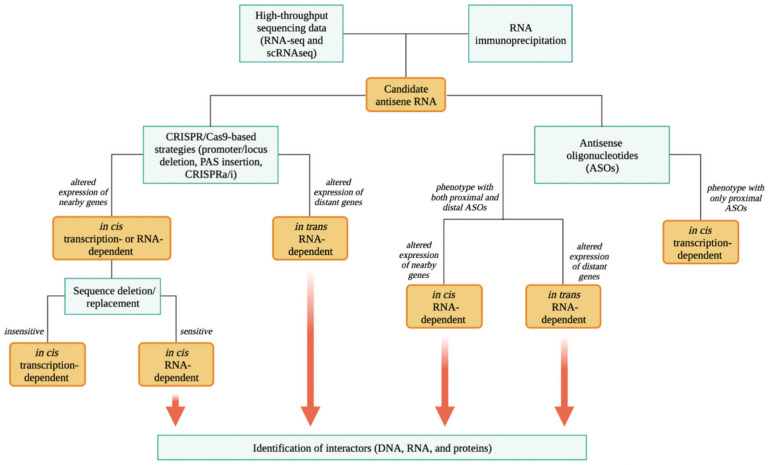
Experimental workflow for asRNA identification and functional study. Techniques and tools are reported in light blue boxes; obtained results are reported in orange boxes. Created with BioRender.com accessed on 25 October 2022.

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
