# Peer review of "Navigating the Multiverse of Antisense RNAs: The Transcription- and RNA-Dependent Dimension"

_ncrna, 2022, doi:10.3390/ncrna8060074_

Round 1

Reviewer 1 Report

The authors have reviewed functions of antisense RNAs with particular emphasis on transcription-dependent and RNA-dependent modes of activity. The review then focuses methodologies commonly utilised to detect natural antisense RNA spices. While significant effort has gone into developing this manuscript, the depth of the review can be improved by adding the following details.

The review lacks information about how antisense RNAs fit into the molecular systems biology of metazoan cells. The two examples below demonstrate how the activity of natural antisense RNAs can be interpreted in a broader context of metazoan cell biology.  

1.     Mosig et al Genes Dev. 2021; 35: 899–913

2.     Vujovic et al Nucleic acids research 2021; 49(18): 10419–10430.

I suggest the authors dedicate a paragraph to discuss this important aspect of NATs before moving to the methodological details.

I also suggest that the authors explain how RNAP-II-dependent transcriptional dynamics that lead to generation of NATs act synergistically with (or complement the activity of) NATs albeit at the DNA level. In other words, transcript-level and transcription-level impacts of NATs must be clearly delineated in the section dedicated to Cis-activities of NATs.

In the following section (methodologies), it could be worthwhile to touch upon methodologies which enable temporally controlled release of synthetic or endogenous NATs.

Reviewer 2 Report

In their review, Pagani et al. provide an overview of non-coding antisense RNAs and their described functions, specifically in eukaryotic cells. They systematically explain the transcription-dependent and -independent functions of asRNAs, distinguishing between overlapping and non-overlapping asRNAs acting through transcription and asRNAs acting through RNA sequence dependent mechanisms in cis and in trans, respectively. They describe a step-by-step workflow for the functional characterization of asRNAs and explain the associated methods. The difficulties in identifying and studying asRNAs are highlighted and discussed. Text and illustration are very well structured and coordinated, making the topic of the review very accessible. The discussion furthermore offers an outlook on the possible therapeutic relevance of asRNAs. Taken together, the review provides a comprehensive and up-to-date summary of the state of research on the subject of asRNA and thus closes a gap in the current literature. I have no major or minor suggestions for improvement and consider the manuscript in its current form to be very complete, informative and up-to-date and therefore suitable for publication.

Author Response

We thank the reviewer for her/his enthusiastic comment on our manuscript.